# Age-period-cohort analysis and projection of cancer mortality in Hong Kong, 1998–2030

Yanji Zhao [iD],[1] Zian Zhuang,[1,2] Lin Yang [iD],[3] Daihai He [iD] [1,4]

YZ and ZZ contributed equally.

[1]Department of Applied Mathematics, The Hong Kong Polytechnic University, Hong Kong, China
[2]Department of Biostatistics, University of California Los Angeles, Los Angeles, California, USA
[3]School of Nursing, The Hong Kong Polytechnic University, Hong Kong, China
[4]Research Institute for Future Food, The Hong Kong Polytechnic University, Hong Kong, China

**Correspondence to**
Dr Daihai He;
daihai.he@polyu.edu.hk

## ABSTRACT

**Objectives** To explore the relationship between immigration groups and cancer mortality, this study aimed to explore age, period, birth cohort effects and effects across genders and immigration groups on mortality rates of lung, pancreatic, colon, liver, prostate and stomach cancers and their projections.

**Design, setting, and participants** Death registry data in Hong Kong between 1998 and 2021, which were stratified by age, sex and immigration status. Immigration status was classified into three groups: locals born in Hong Kong, long-stay immigrants and short-stay immigrants.

**Methods** Age-period-cohort (APC) analysis was used to examine age, period, and birth cohort effects for genders and immigration groups from 1998 to 2021. Bayesian APC models were applied to predict the mortality rates from 2022 to 2030.

**Results** Short-stay immigrants revealed pronounced fluctuations of mortality rates by age and of relative risks by cohort and period effects for six types of cancers than those of long-stay immigrants and locals. Immigrants for each type of cancer and gender will be at a higher mortality risk than locals. After 2021, decreasing trends (p<0.05) or plateau (p>0.05) of forecasting mortality rates of cancers occur for all immigration groups, except for increasing trends for short-stay male immigrants with colon cancer (p<0.05, Avg+0.30 deaths/100 000 per annum from 15.47 to 18.50 deaths/100 000) and long-stay male immigrants with pancreatic cancer (p<0.05, Avg+0.72 deaths/100 000 per annum from 16.30 to 23.49 deaths/100 000).

**Conclusions** Findings underscore the effect of gender and immigration status in Hong Kong on mortality risks of cancers that immigrants for each type of cancer and gender will be at a higher mortality risk than locals.

## INTRODUCTION

Several migration waves from mainland China to Hong Kong have occurred over the past century. These migration waves included a large-scale migration inflow from 1945 to 1950 (the Chinese Civil War) and a few small-scale inflows in the 1950s, 1970s and 1990s.[1–3] In 2016, immigrants from mainland China formed approximately 38% of the population of Hong Kong. These inflows have led to a growing interest in research on the disparity

### STRENGTHS AND LIMITATIONS OF THIS STUDY

⇒ This study provides new evidence regarding the relationship between immigration status and cancer mortality, given the effects of age, period, birth cohort and their predictions.
⇒ The non-identifiability problem has not been interpreted in age-period-cohort models.
⇒ The future perspective of cancer therapies and techniques has not been considered.

of health conditions between the locals and immigrants.

Cancer has been one of the most common causes of death, as an estimated 19.3 million new cancer cases and 9.9 million new cancer-associated deaths occurred worldwide in 2020.[4] In Hong Kong, lung cancer is one of the most common causes of cancer deaths.[5 6] Previous studies suggested that the primary cause of lung cancer is cigarette smoking.[7–11] Genetic factors, asbestos, radon gas, second-hand smoke and other forms of air pollution have been proven to influence the risk of lung cancer.[12–18] The overall daily smoking rate in mainland China was approximately 23.2% in 2018,[19] whereas the daily smoking rate in Hong Kong was only 10.2% in 2019.[20] The leading causes of liver cancer include viral infection, drinking of alcohol and polluted water and food supplies which are also culprits for colon, stomach and pancreatic cancer.[21] Alcohol consumption per capita in Hong Kong has reached 2.37 L in 2021,[22] compared with 7.0 L of per capita consumption of alcohol in mainland China in 2018.[23] As approximately 99% of prostate cancer cases occur after age 50, factors of prostate cancer have been regarded as old age, race, family history and the diet of red meat consumption.[24] In addition to these risk factors, studies have suggested that cancer mortality rates vary depending on migrant status.[25–28] According to data from the Census and Statistics Department of Hong Kong, approximately 81% of immigrants in Hong

Kong immigrated from mainland China, Macau and Taiwan. Immigrants from mainland China account for the bulk of this population. Previous studies have shown that child immigrants in Hong Kong tend to suffer from a higher risk of wheezing disorders and cardiovascular diseases, and immigrant women have higher age-specific mortality rates of breast cancer than locally born women in Hong Kong.[29 30] However, to date, few studies have investigated the effect of length of stay in Hong Kong and birthplace on the risk of other types of cancer.

In this study, we compared the mortality rates of lung, pancreatic, colon, liver, prostate and stomach cancers between locally born residents in Hong Kong and immigrants from mainland China. Both populations are widely considered as ethnically homogeneous with similar cultures. Nevertheless, due to different early life experiences, immigrants are exposed to more various social economy and lifestyles than locals. Therefore, it is constructive to ascertain whether immigrants from mainland China have a different mortality pattern of cancers from locals to verify the significance of migration status for this health outcome. As age-period-cohort (APC) analysis plays a vital role in studying time-specific phenomena in epidemiology, in this study, to evaluate the effect of immigration on cancer mortality in the past and future, we developed APC models specified by sex and migrant status to assess the effects of age, period, birth cohort and of the length of stay in Hong Kong on the mortality risks of cancers. Additionally, we explore the projection of mortality rates for the locally born population and immigrants in Hong Kong who were younger or older than 60 using a predictive model, taking into account age, period and birth cohort effects as well.

## METHODS
### Data
We obtained the death registry data in Hong Kong between 1998 and 2021 from the Census and Statistics Department of Hong Kong, as the data in 2022 has not been available up to now. The data was extracted from a routine census held by the Hong Kong government as subjective errors caused by resampling can be neglected. The population data were stratified by age, sex, immigration status and length of stay in Hong Kong. We retrieved six types of cancer cases from the death registry data using ICD codes, such as ICD-9 code 162 and ICD-10 codes C34.0–C34.3, C348 and C349 for lung cancer. To assure comparability among registries, deaths from the age group of 35–85 years were selected, since cases younger than 35 and older than 85 were relatively trivial for lack of statistical interpretability.[31] Immigration status was classified into three groups: locals born in Hong Kong, immigrants who have lived in Hong Kong for >10 years before death defined as long-stay immigrants, and immigrants who have lived in Hong Kong for ≤10 years before death defined as short-stay immigrants. Notably, much focus was placed on immigrants from mainland China, because approximately 81%

of immigrants in Hong Kong came from mainland China, Macau and Taiwan based on the data from the Census and Statistics Department of Hong Kong. Moreover, few cases recorded from Macau and Taiwan are statistically insignificant in the analysis. Demographics and population projections from 2022 to 2030 were retrieved from the Census and Statistics Department of Hong Kong and estimated with cubic smoothing spline as the prerequisite of the predictive model. Codes for APC and Bayesian age-period-cohort (BAPC) analysis are available in the GitHub repository (https://github.com/kshz2164313/APC-population-projections-for-immigration-HK).

### Statistical analysis
We modelled cancer mortality rates in Hong Kong using APC analysis based on log-linear Poisson regression models. The model aimed to disentangle age, period and cohort effects of time-varying phenomena simultaneously,[32 33] given that

$$\log(E_{ij}) = \alpha_i + \beta_j + \gamma_k + \mu + \log(\theta_{ij}) \quad (1)$$

where $E_{ij}$ denotes expected mortality; $\alpha_i$, $\beta_j$, $\gamma_k$ denote age, period and cohort effect, respectively, for $i$=1, …, I, $j$=1, …, J, $k$=1, …, K with $k=I-i+j$. $\log(\theta_{ij})$ is the offset. We mainly focused on the contributions of sex and immigration status due to the non-identifiability problem that the effects of these three components are collinear with each other (denoted as period–age=cohort).[34] Birth cohort effect and period effect were assessed with relative risks to evaluate the effect of three components. The median year of birth among cases was regarded as the reference cohort.[35 36] Since death cases aged 35–85 years between 1998 and 2021 were selected, the range of birth cohort from 1913 to 1986 covered observations and further projections until 2030. The second and penultimate period effects were constrained to the reference for period. For sex and immigration status, maximum likelihood framework was applied to estimate the relative risks and 95% CIs by age groups, calendar period and birth cohort.

Several projection approaches for future cancer mortality have been developed, but a BAPC model built on integrated nested Laplace approximations[37] yields relatively higher coverage and better performance for all evaluated parameter combinations.[38] To prevent some sampling problems caused by Markov chain Monte Carlo (MCMC), this MCMC-free BAPC approach was applied to predict future cancer mortality within a fully Bayesian inference setting and provide outputs of interest simply, such as projected age-standardised and age-specific rates. Convergence checks are not necessary for this technique.[37] The projections of age-standardised cancer mortality rates for each sex, age group (younger or older than 60 years) and migrant status, taking into account age, period and birth cohort effects, were performed based on the weights of population age groups from the WHO World Standard population,[39] with 95% prediction intervals. The Mann-Kendall trend test was applied to

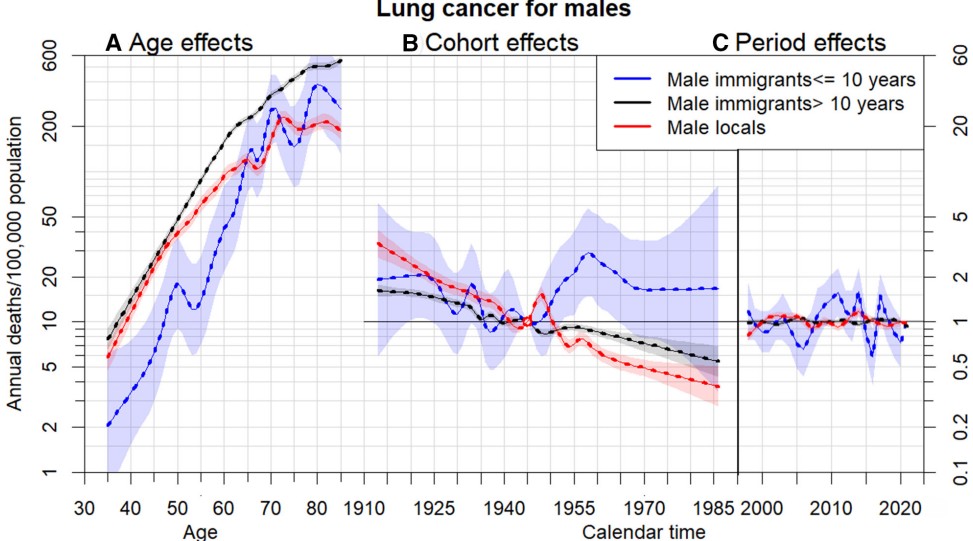

**Figure 1** Parameter estimates of age (A), cohort (B) and period (C) effects based on an age-period-cohort model of male lung cancer mortality rates by immigrant groups: locals, immigrants have stayed in Hong Kong for more than 10 years and immigrants have stayed in Hong Kong for less than or equal to 10 years. Age effect was assessed by mortality (left axis). Cohort and period effects were assessed by relative risk (right axis), 95% CIs are shown as shaded bands.

verify the projection trend. Friedman's two-way analysis of variance was applied to test interactions between gender and immigration groups for each year.

All analyses were performed via R V.4.2.1 (R Core Team, R Foundation for Statistical Computing, Vienna, Austria, 2013, http://www.R-project.org/). The APC models were established using the Epi package, and the projections based on Bayesian APC models were performed with the BAPC package.

**Patient and public involvement**
None.

## RESULTS

Figures 1 and 2 and online supplemental eFigure 1A–E the estimates of age (assessed by cancer mortality), cohort and period effects (assessed by relative risk) based on APC models among three migrant groups for men and women with six types of cancers, respectively. All the mortality rates for each gender and immigration status exhibit notable increasing trends with age. Age, cohort and period effects of six types of cancer for immigrants who stayed in Hong Kong for ≤10 years revealed relatively more pronounced fluctuations and deviations from

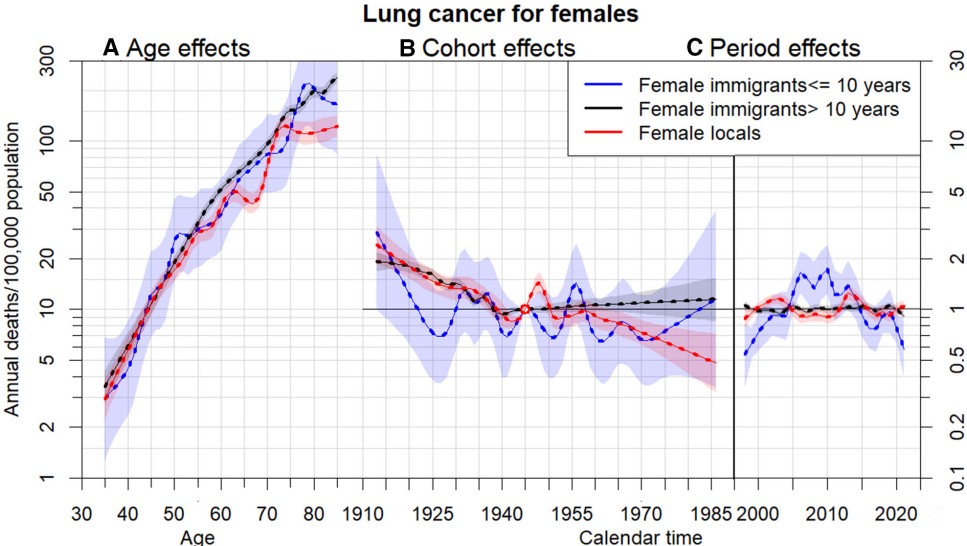

**Figure 2** Parameter estimates of age (A), cohort (B) and period (C) effects based on an age-period-cohort model of female lung cancer mortality rates by immigrant groups: locals, immigrants have stayed in Hong Kong for more than 10 years and immigrants have stayed in Hong Kong for less than or equal to 10 years. Age effect was assessed by mortality (left axis). Cohort and period effects were assessed by relative risk (right axis), 95% CIs are shown as shaded bands.

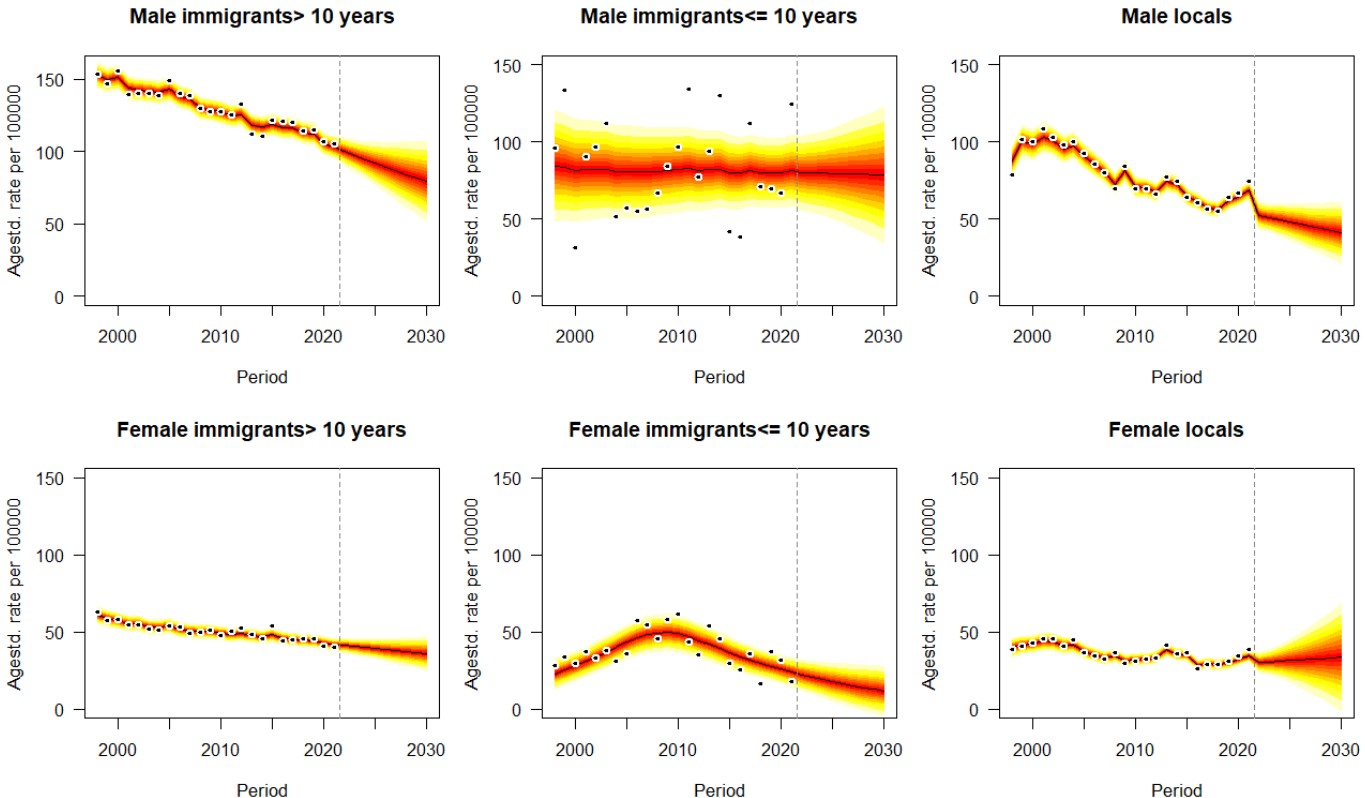

**Figure 3** Projections of lung cancer mortality rates by gender and immigrant status from 2022 to 2030. Observations are shown as dots with the predictive distribution between the 5% and 95% quantile, whereby each lighter shade of red represents an additional 10% predictive CI. The predictive mean is shown as black solid line and the vertical dashed line indicates where prediction started.

those effects in the other two immigration groups. Significant increasing trends of age effect occurred in all types of cancer, regardless of gender and immigration status. For example, while relatively insignificant differences in lung cancer mortality rates by immigration status among females have performed, male immigrants who remained in Hong Kong for >10 years had higher lung cancer mortality rates at ages above 50 years and those who arrived ≤10 years had lower lung cancer mortality at ages below 62 years compared with local men (figure 1). In addition to compatible dynamics of period effect for locals and long-stay immigrants, similar changes of relative risks by birth cohort for locals and long-stay immigrants in lung, colon, liver and stomach cancers occurred before 1945, whereas significant differences of relative risks by birth cohort between these two immigration groups occurred after 1960 (figure 1 and online supplemental eFigure 1A,B,D). Locals and long-stay immigrants in pancreatic and prostate cancer perform almost similar changes of relative risks by birth cohort effects all the time (online supplemental eFigure 1C,E). Short-stay immigrants who have stayed in Hong Kong for ≤10 years had more fluctuating relative risks affected by period effects before 2020 than those for locals and long-stay immigrants. Lack of young cases, especially young short-stay immigrants, of prostate cancer leads to significant deviations and variances in age and cohort effects.

Figures 3–5, online supplemental eFigures 2–6 the age-standardised mortality rates of six types of cancer from 1998 to 2021 and their projections by sex, immigrant status and age groups from 2022 to 2030, taking into account age, period and birth cohort effects. Means and SD of predictive mortality rates are shown in online supplemental eTables 1–6. For all ages projection (figure 2 and online supplemental eFigure 2–6), as approximately significant interactions between gender and immigration groups emerge for each type of cancer in each year (p<0.05), given the projected trends, immigrants for each gender, especially who have stayed in Hong Kong for >10 years will suffer from higher mortality rates of cancer in each year than locals. Monotone decreasing trends or plateau of forecasting occur for both genders and all immigration groups in cancers, except for increasing trends for male immigrants who have stayed in Hong Kong for ≤10 years with colon cancer (p<0.05, Avg+0.30 deaths/100 000 per annum) from 15.47 deaths/100 000 (95% CI 11.28, 19.66) in 2021 to 18.50 deaths/100 000 (95% CI 2.31, 34.69) in 2030, and male immigrants who have stayed in Hong Kong for >10 years with pancreatic cancer (p<0.05, Avg+0.72 deaths/100 000 per annum) from 16.30 deaths/100 000 (95% CI 14.38, 17.26) in 2021 to 23.49 deaths/100 000 (95% CI 12.49, 34.49) in 2030. Most of predictive trends for

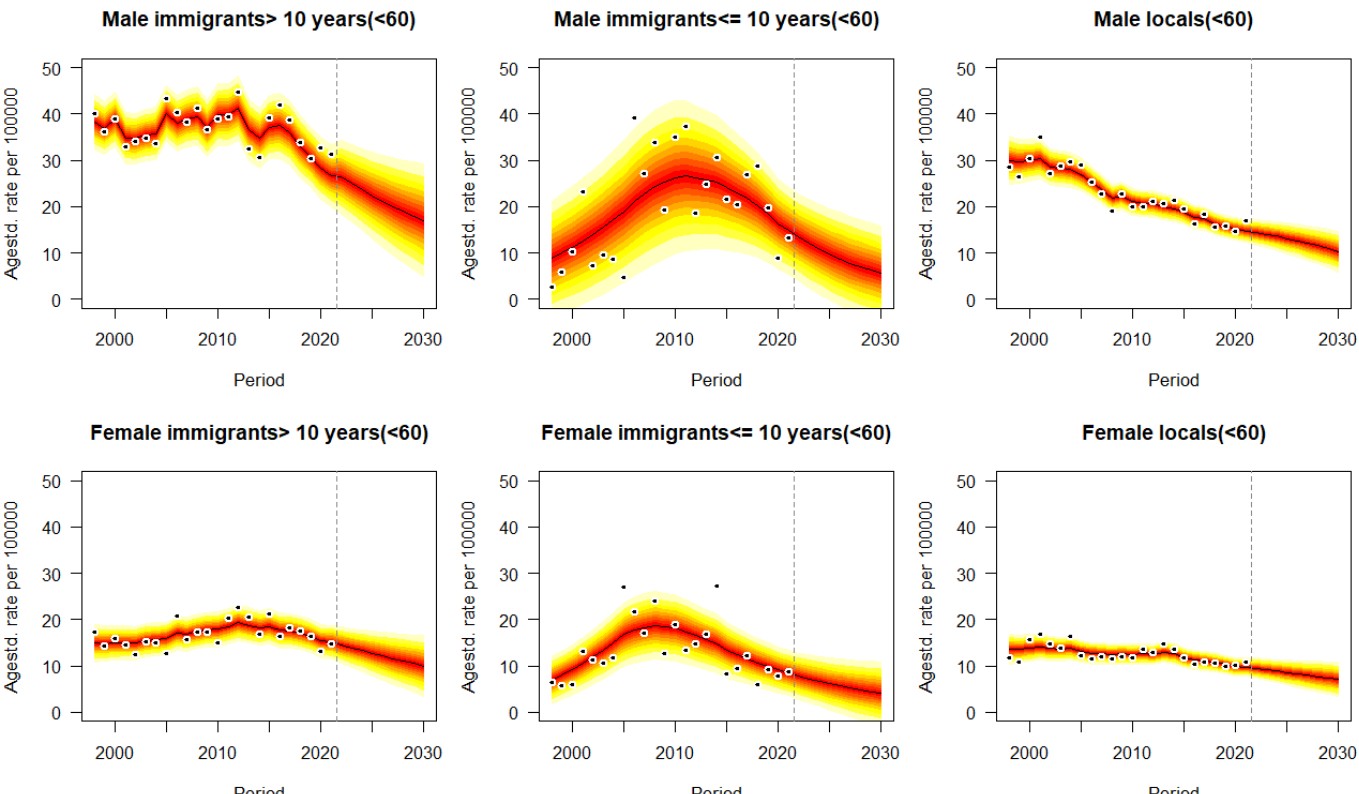

**Figure 4** Projections of lung cancer mortality rates for the population younger than 60 by gender and immigrant status from 2022 to 2030. Observations are shown as dots with the predictive distribution between the 5% and 95% quantile, whereby each lighter shade of red represents an additional 10% predictive CI. The predictive mean is shown as black solid line and the vertical dashed line indicates where prediction started.

younger cases (<60 years) and older cases (≥60 years) reach a consensus with those for all ages population, except for two phenomena: (1) mortality rates of lung cancer for men immigrants ≤10 that insignificant trend for all ages (p>0.05) versus decline for younger cases (p<0.05) versus increase for older cases (p<0.05); (2) mortality rates of liver cancer for men immigrants >10 that decline for all ages (p<0.05) versus decline for younger cases (p<0.05) versus insignificant trend for older cases (p>0.05). Some particular cases occur in the projection of prostate cancer that young long-stay male immigrants (0.44 deaths/100 000, 95% CI 0, 1.05) aged less than 60 will be at lower mortality rate than locals (0.69 deaths/100 000, 95% CI 0, 1.42) in 2030 (online supplemental eTable 6). Compared with other cancers and immigration groups, male immigrants who have stayed in Hong Kong for >10 years with lung cancer would perform the most significant decline in predictive mean from 102.90 (95% CI 98.14, 107.66) to 79.55 (95% CI 47.46, 111.64) deaths per 100 000 population (Avg −2.34 deaths/100 000 per annum) (online supplemental eTable 1), while the same immigration group with pancreatic cancer would indicate the most significant uptrend in each year of 16.30 (95% CI 14.38, 17.26) and 23.49 (95% CI 12.49, 34.49) deaths per 100 000 population in

2021 and 2030, respectively (Avg+0.72 deaths/100 000 per annum) (online supplemental eTable 4).

## DISCUSSION

Early detection of cancer is positive and instructive for increasing chances of cure. Nevertheless, the high mortality rate of cancer results from late diagnosis among most patients after progression to more advanced or severe stages. Individuals at high risk of cancer, such as smokers, alcoholics or those who are frequently exposed to susceptible circumstances, should be screened for early detections to increase opportunities for cure.[40] Therefore, the differences in mortality rates among immigration groups are synonymous with detection means, therapies, and social history in disparate periods and areas.

While the changes in mortality rates by age for long-stay immigrants reached approximate harmony with those for locals, the changes in mortality rates by age for short-stay immigrants revealed clear differences with those for the other two populations. The group of long-stay immigrants had a higher risk of death from lung, colon and liver cancers than the other two immigration groups after the age of 60 years. Short-stay male immigrants were less likely to die from lung cancer before the age of 65 years. The contrast in age effects among the immigration groups

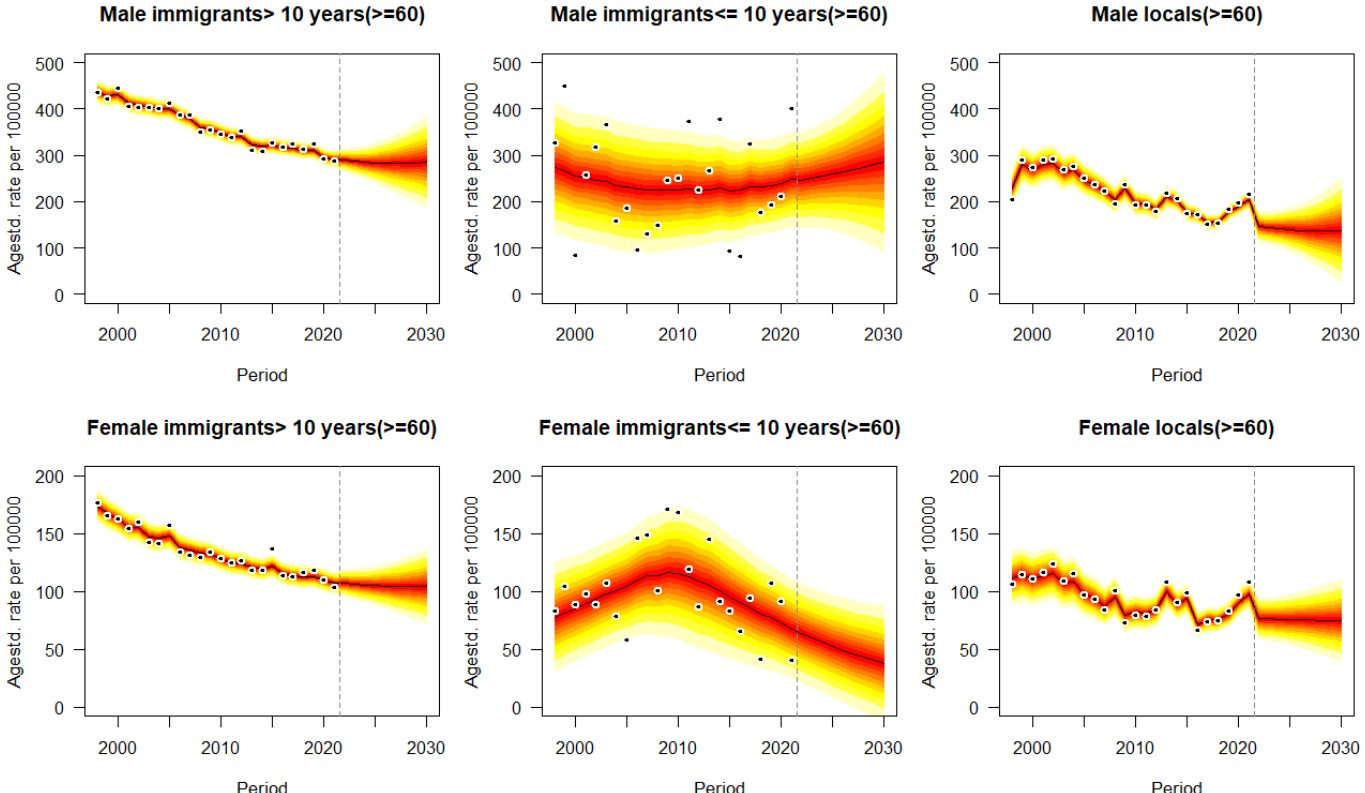

**Figure 5** Projections of lung cancer mortality rates for the population older than or equal to 60 by gender and immigrant status from 2022 to 2030. Observations are shown as dots with the predictive distribution between the 5% and 95% quantile, whereby each lighter shade of red represents an additional 10% predictive CI. The predictive mean is shown as black solid line and the vertical dashed line indicates where prediction started.

was partially consistent with studies[25 41] that highlighted the age effects for locals and immigrants on breast cancer mortality in Hong Kong and lung cancer incidence in Sweden, as they both showed similar trends and magnitudes between locals and immigrants before the age of 60 years. They are also compatible with the results in[42] that diagnosis of liver cancer is the most frequent among populations at 55–65 years old. According to these trends, young individuals, especially new young immigrant men, who have benefited from all-rounded development in mainland China and Hong Kong, are more likely to seek early detection and be treated for cancers using more advanced treatments.[43] Differences in birth cohort effects among immigrant groups partially comply with the interpretation above.

We observed significant trends of cohort effects among locals and immigrants. These findings are partially consistent but subtly different from previous findings, regarding the effect of immigration status on cancers. Zhao *et al*[25] described multiple peaks of cohort effects on breast cancer mortality between locals and immigrants in Hong Kong, as well as a significant decline in cohort effects after 1950. In contrast, Sung *et al*[44] investigated the difference in breast cancer incidence between Chinese Americans and non-Hispanic whites in the USA and emphasised that Chinese Americans were at lower risk of breast cancer than non-Hispanic whites born in the same year. Here,

we interpret the cohort-driven trends resulting from the intricacy of social history and lifestyle. Compared with a relatively stable social development in Hong Kong, representing downward trends of relative risks for locals, wars and social instability in mainland China resulted in several immigration waves from mainland China to Hong Kong before 1950. Additionally, remarkable increasing trends were recorded for new immigrants after 1950, which corresponded to the economic downturn after wars and famine between 1959 and 1961 during their youth.[45] The increasing trends for new immigrants and similar trends for locals and long-stay immigrants were consistent with the finding that nutrient deficiency contributes to a higher risk of severe mortality rates of cancers.[46] Furthermore, we speculate that these trends, especially those for locals and long-stay immigrants, are most likely attributed to social development and personal behaviours, such as daily habits, occupational history, different diagnoses and treatments, and domestic environmental exposures. Notably, short-stay immigrants suffered from a lower risk of death from colon cancer for all ages (online supplemental eFigure 1A). As locals and immigrants in Hong Kong transitioned to more westernised lifestyles, higher consumption of meat was associated with a higher risk of these types of cancer, whereas consumption of vegetables had a strong protective effect against pancreatic cancer, and moderate consumption of coffee appeared

to be beneficial against lung cancer.[6 47] Further studies on potential risk factors are required.

Short-stay immigrants had more fluctuating and non-stationary but inconspicuous relative risks by period effects before 2021 than locals and long-stay immigrants. Cumulatively, an arch pattern and fluctuating curve depicting period effects externally resulted in an arch pattern of age-standardised mortality rates for short-stay immigrant women and irregular rates for short-stay immigrant men before 2021. The external performance of different period effects on mortality rates could be most likely attributed to the higher effect of different lifestyles and social development on new immigrants than on long-stay immigrants and locals in Hong Kong. For the age-standardised mortality rates and projections, consistent with previous findings,[48 49] we predict that the mortality rates of cancer in Hong Kong after 2021 will continue to decline or remain relatively stable, consistent with the trends before 2020, except for male immigrants who have stayed in Hong Kong for ≤10 years with colon cancer and male immigrants who have stayed in Hong Kong for >10 years with pancreatic cancer. Men will be at higher risk of mortality rates of cancer than women, regardless of immigration status. They are also compatible with the results in[4] that men suffer from a higher risk of these types of cancer than women, excluding prostate cancer. Furthermore, new immigrant women will be at lower risk than local women, even though long-stay immigrants will suffer from higher mortality rates than locals in the future. Potential interpretations could be consistent with those for birth cohort effects, as age and period effects are considered as confounders of cohort effects.

In the past few decades, spurred by an increasing burden of high incidence and mortality rates of cancer, several studies focused on the inherent identification dilemma of three effects in the APC model. Further, complicated population distribution and immigration status in Hong Kong, one of the areas with the highest population density and migration frequency in the world, have intricate causes and inherent dynamics of cancer and other diseases. To our knowledge, few studies have assessed the relationship between immigration status and cancer mortality. Therefore, this study is original to examine the effect of the length of stay in Hong Kong and origins of previous residence on cancer deaths, which is instructive for further immigration policy-making and targeted strategies of disease detection and intervention. However, this study had several limitations. Given the non-identifiability problem in APC models, we could only depict trends and variations among different immigration and sex groups, as illustrated in figures, and insufficiently perform the estimates of the contributions of three effects or subgroups to mortality rates. Furthermore, we adopted a cubic smoothing spline to estimate populations of immigrants and locals due to the large proportion of unspecified immigration status from official demographic projections. A few acceptable cases resulted in a limited type of cancer so that some common cancers, such as the ovary and cervix, were discarded. Since the issue of quantification, the future perspective of cancer therapies and techniques has not been considered in the model of projection.

## CONCLUSION

We conclude that immigrants, especially short-stay immigrants, had more pronounced fluctuations of mortality rates by age and of relative risks by cohort and period effects for six types of cancers than those of long-stay immigrants and locals. Male immigrants who have stayed in Hong Kong for ≤10 years with colon cancer and male immigrants who have stayed in Hong Kong for >10 years with pancreatic cancer would perform significant uptrend in the future, while other immigration groups for each type of cancer would continue to decline or remain relatively stable. Immigrants for each gender in Hong Kong would suffer from higher mortality risks of cancers than locals in the future.

**Correction notice** This article has been corrected since it was first published. Affiliations have been updated.

**Contributors** YZ: methodology, formal analysis, data curation, writing - original draft, writing - review and editing, visualisation, guarantor. ZZ: methodology, formal analysis, data curation, writing - review and editing. LY: validation, writing - review and editing. DH: conceptualisation, writing - review and editing, supervision.

**Funding** The work described in this paper was partially supported by a grant from the Research Grants Council of the Hong Kong Special Administrative Region, China (HKU C7123-20G).

**Competing interests** None declared.

**Patient and public involvement** Patients and/or the public were not involved in the design, or conduct, or reporting, or dissemination plans of this research.

**Patient consent for publication** Not applicable.

**Ethics approval** Not applicable.

**Provenance and peer review** Not commissioned; externally peer reviewed.

**Data availability statement** Data are available upon reasonable request.

**ORCID iDs**
Yanji Zhao http://orcid.org/0000-0002-2992-0164
Lin Yang http://orcid.org/0000-0002-5964-3233
Daihai He http://orcid.org/0000-0003-3253-654X

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
