## [Reviewer comments · BMJ Open]

ARTICLE DETAILS

TITLE (PROVISIONAL)	An age-period-cohort analysis and projection of cancer mortality in Hong Kong, 1998–2030
AUTHORS	Zhao, Yanji; Zhuang, Zian; Yang, Lin; He, Daihai

VERSION 1 – REVIEW

REVIEWER	Rosenberg, Philip NIH, Bethesda, Biostatistics Branch
REVIEW RETURNED	17-Apr-2023

GENERAL COMMENTS	This is an interesting comparative analysis and the methods are nicely executed. The graphs are particularly informative. The main limitation is strong changes in birth cohort trends are apparent for many cancers and populations. This implies that the trend in mortality over time necessarily vary by age. Given the cancers included in the study, an additional focus on 'early onset' cases ≤ 60, versus older cases $60+$, would greatly increase the generalizability of the conclusions. The authors demonstrate mastery of the Bayesian Age-Period-Cohort forecasting model. My understanding is these models are producing mortality forecasts over time periods within age groups. Hence, it should be straight-forward for the authors to calculate age-standardized forecasts within the two age groups suggest above. These data could be added to the tables (including the eTables) to the overall forecasts shown in the current version. A summary figure showing observed and forecast age-standardized mortality over time by age group by site and sex - this would add a great deal of valuable information. Then, the two main conclusions of the paper could be extended to each age group, namely: 1) "Men will be at a higher risk of mortality from cancers than women in the future, (except for prostate cancer)" Is this equally so for early-onset cases? and 2) Long-stay immigrants for each type of cancer and gender will be at a higher mortality risk than locals. Again - is this equally so for early-onset cases?
--

	A minor point: "Immigrants had more pronounced fluctuations and sharper slopes of age, cohort, and period effects than locals" - I didn't spot how this was quantified.
--	---

REVIEWER	Kane, Eleanor University of York, Health Sciences
-----------------	--

REVIEW RETURNED	29-Apr-2023
-------------

GENERAL COMMENTS	The study examines changes in mortality rates from lung, pancreatic, colon, liver, prostate and stomach cancers among immigration groups compared to the local population in Hong Kong. Using death registry data 1998-2021, age-period-birth cohort effects were examined among short- and long-stay immigrants; and predictions of mortality rates for 2022-2030 were estimated. Over time, mortality rates of colon cancer and pancreatic cancer increased among male short-term and long-term immigrants respectively. Lung cancer death rates among male long-term immigrants were predicted to decrease the most, while pancreatic cancer deaths rates in the same group were predicted to have the greatest increase. Abstract Needs to be clearer. Objectives do not specify why examining- and predicting- trends in cancer mortality among immigrants in Hong Kong is of interest. Results should describe the key findings- currently difficult to understand- adding some key statistics to illustrate would help. Conclusions need to tie with findings (sex differences are stated but not described in the abstract results; prostate cancer is highlighted but can only occur in men); long-stay immigrants are at higher risk of cancer mortality than local population but again this is not clear in the abstract results. Strengths and limitations of the study The text here does not make sense. Please reconsider this section Lines 147-150 "The median dates of birth among cases were regarded as the reference cohort, while the second and penultimate period effects were constrained to the reference groups, as birth cohort effect and period effect were assessed with relative risks"- Unless there is some detail missing in this sentence, there can only be one median date of birth among cases. The other part of the sentence is unclear. For future projections, what assumptions were made for birth cohorts who were not present in the data but who would be in future populations? Graphs are small and as such, are difficult to see; this is particularly so for the period graph where the x-axis should only extend to 2021 rather than 2040. Please check the titles are consistent with the graph headers (cohort and period effects are labelled differently in titles from graph headers). Most are labelled as "morality" instead of "mortality". In the projection graphs, the black line representing the predictive mean is difficult to see- the authors may want to consider a color of other than blue- or a different shade- for the 5-95% quantiles. Changing the aspect ratio of the projection graphs (i.e. make bit taller) may also help.
--

	Lines 183-185 “For example, compared to no significant effects of immigration status for women on mortality rates of lung cancer by age (Figure 1b), the higher age effect for men who have stayed in Hong Kong for >10 years occurred after the age of 50 years and the lower age effect of men who had short stays occurred before the age of 62 years (Figure 1a).” Note that men and women are not being compared as such- it would be better to say that while there was no difference in lung cancer mortality rates by immigration status among women, male immigrants who remained in Hong Kong for >10 years had higher lung cancer mortality rates at ages above 50 years and those who arrived ≤10 years had lower lung cancer mortality at ages below 62 years compared to local men. Other descriptions should follow this suggestion, making reference to the specific cancer mortality rates too. Line 205- the text implies that prostate cancer could occur in either sex- please modify the text. Similarly in line 283. As rates are presented by sex, why did the authors not consider reporting patterns for the most common cancers for each sex- so for women, breast, ovary, cervix? Lines 227-8 Cancer mortality rates increasing with age is known- what is relevant here is whether patterns are different among the immigrant and local populations. Consider removing this first sentence. Please check that the conclusions focus on the study objectives (differences in cancer mortality rates in different immigrant and local populations by age, cohort and period) and are consistent with the reported results. Please check the ethical statement- although the study uses routinely collected data, the study does involve human participants so the ethical statement may warrant revision.
--	---

REVIEWER	Yano, Yoshitaka Kyoto Pharmaceutical University, Education and Research
REVIEW RETURNED	20-May-2023

GENERAL COMMENTS	This manuscript describes the results of mathematical data analysis for cancer mortality data in Hong Kong, with some prediction of future trends. The mathematical model is modified based on the previously reported models based on the Poisson model. I, as a reviewer, think the data analysis for the profiles is carried out accordingly, however some discussions regarding the reasons of the profile changes based on the progress of cancer therapy would be required. Here are some comments.  1. It is not clear that the data are full-size data, I mean the number of deaths is a complete survey in the country. If the data were based on spontaneous report, it may make some biases in the analysis. 2. Is not it necessary to describe the statistical significance of the model parameters. As the descriptions of the model development process are limited in this manuscript - mostly the manuscript is just quoting other references, I could not completely understand the parameter handling. 3. For example, a sentence “... perform the most significant decline (Page 2, Line43), ... most significant uptrend (Page 2, Line
--

	45) ...”, seems subjective and not based on strict mathematical basis. 4. The future profile prediction is only based on the available data until 2022 with the covariate of age, cohort only. In fact, recent cancer therapy is progressing remarkably, and the future perspective of such progresses should be considered for the prediction. I think this is one limitation of the applied model here.
--	---

VERSION 1 – AUTHOR RESPONSE

Comments from the Reviewer #1, and author’s response

Comment1: The main limitation is strong changes in birth cohort trends are apparent for many cancers and populations. This implies that the trend in mortality over time necessarily vary by age.

Given the cancers included in the study, an additional focus on 'early onset' cases ≤ 60 , versus older cases $60+$, would greatly increase the generalizability of the conclusions.

The authors demonstrate mastery of the Bayesian Age-Period-Cohort forecasting model. My understanding is these models are producing mortality forecasts over time periods within age groups. Hence, it should be straight-forward for the authors to calculate age-standardized forecasts within the two age groups suggest above.

These data could be added to the tables (including the eTables) to the overall forecasts shown in the current version.

A summary figure showing observed and forecast age-standardized mortality over time by age group by site and sex - this would add a great deal of valuable information.

Thank you very much for your comments! The trend in mortality over time did vary by age shown from the age effect on the figures of APC models. Based on our dataset, we consider the early onset cases who were younger than 60 as well as older cases ≥ 60 years old. Two new projection figures and a revised table with two age groups for each type of cancer are presented in both main text (Figure 1-4) and Supplementary materials (eFigure 1-6 and eTable 1-6). Some statements were modified as follows:

Line 155-156

“The projections of age-standardized cancer mortality rates for each sex, age group (younger or older than 60 years) and migrant status, taking into account age, period, and birth cohort effects, were performed”

Comment 2: Then, the two main conclusions of the paper could be extended to each age group, namely:

1) "Men will be at a higher risk of mortality from cancers than women in the future, (except for prostate cancer)"

Is this equally so for early-onset cases?

Yes. We updated our statement as follows:

Line 208

“Men will be at higher risk of mortality rates of cancers (excluding prostate cancer) than women in the future for all three age groups (all ages, young and older than 60 years).”

Comment 3: and 2) Long-stay immigrants for each type of cancer and gender will be at a higher mortality risk than locals.

Again - is this equally so for early-onset cases?

Results have been changed when we considered early-onset cases, and we revised them as follows:
Line 210-216

“Given the future developing trends, immigrants, especially the group of immigrant who have stayed in Hong Kong for >10 years with lung, liver, pancreatic, prostate and colon cancer, will have relatively higher mortality rates in each year for each gender compared with locals and pronounced decline in predictive means (all p-values < 0.05). Some particular cases occur in the projection of prostate cancer that young long-stay male immigrants (0.44 deaths/100,000) aged less than 60 will be at lower mortality rate than locals (0.69 deaths/100,000) in 2030 (eTable 6).”

Comment 4: A minor point: "Immigrants had more pronounced fluctuations and sharper slopes of age, cohort, and period effects than locals" - I didn't spot how this was quantified.

We applied a more clear statement in “Results” as follows:

Line 37-39

“Short-stay immigrants revealed pronounced fluctuations of mortality rate by age and of relative risks by cohort and period effects for six types of cancers than those of long-stay immigrants and locals.”

And Line 183-190

“similar change of relative risks by birth cohort for locals and long-stay immigrants in lung, colon, liver and stomach cancers occurred before 1945, whereas significant differences of relative risks by birth cohort between these two immigration groups occurred after 1960. Locals and long-stay immigrants in pancreatic and prostate cancer perform almost similar change of relative risks by birth cohort effects all the time. Short-stay immigrants who have stayed in Hong Kong for ≤10 years had more fluctuating relative risks affected by period effects before 2020 than those for locals and long-stay immigrants.”

Comments from the Reviewer #2, and author's response

Comment 1: The study examines changes in mortality rates from lung, pancreatic, colon, liver, prostate and stomach cancers among immigration groups compared to the local population in Hong Kong. Using death registry data 1998-2021, age-period-birth cohort effects were examined among short- and long-stay immigrants; and predictions of mortality rates for 2022-2030 were estimated. Over time, mortality rates of colon cancer and pancreatic cancer increased among male short-term and long-term immigrants respectively. Lung cancer death rates among male long-term immigrants were predicted to decrease the most, while pancreatic cancer deaths rates in the same group were predicted to have the greatest increase.

Abstract

Needs to be clearer. Objectives do not specify why examining- and predicting- trends in cancer mortality among immigrants in Hong Kong is of interest. Results should describe the key findings- currently difficult to understand- adding some key statistics to illustrate would help. Conclusions need to tie with findings (sex differences are stated but not described in the abstract results; prostate

cancer is highlighted but can only occur in men); long-stay immigrants are at higher risk of cancer mortality than local population but again this is not clear in the abstract results.

Thank you very much for your comments! We have improved our abstract: specified the motivation of the research; updated results as well as Mann-Kendall trend test to support our conclusions as follows:

Line 24-27

“Objectives: Complicated population distribution and immigration status in Hong Kong have brought out intricate causes of diseases. This study was aimed to explore age, period, birth cohort effects and effects across genders and immigration groups on mortality rates of lung, pancreatic, colon, liver, prostate and stomach cancers and their projections.”

Line 37-45

“Results: Short-stay immigrants revealed pronounced fluctuations of mortality rates by age and of relative risks by cohort and period effects for six types of cancers than those of long-stay immigrants and locals. Decreasing trends ($p < 0.05$) or plateau ($p > 0.05$) of forecasting mortality rates of cancers occur for all immigration groups, except for increasing trends for short-stay male immigrants with colon cancer (16.77 deaths to 18.50 deaths/100,000 $p < 0.05$) and long-stay male immigrants with pancreatic cancer (17.87 deaths to 23.49 deaths/100,000 $p < 0.05$). Men will be at a higher risk of mortality from cancers than women in the future (excluding prostate cancer). Immigrants for each type of cancer and gender will be at a higher mortality risk than locals.”

Line 47-49

“Conclusions: Findings underscore the effect of gender and immigration status in Hong Kong on mortality risks of cancers that immigrants for each type of cancer and gender will be at a higher mortality risk than locals.”

Comment 2: Strengths and limitations of the study

The text here does not make sense. Please reconsider this section

Thank you for your comment. This part is required from the journal and we revised this part based on editor's comment to make them clear on page 3 as follows:

Line 55-60

“This study provides new evidence regarding the relationship between immigration status and cancer mortality, given the effects of age, period, birth cohort and their predictions.

The non-identifiability problem has not been interpreted in APC models

The future perspective of cancer therapies and techniques have not been considered.”

Comment 3: Lines 147-150 “The median dates of birth among cases were regarded as the reference cohort, while the second and penultimate period effects were constrained to the reference groups, as birth cohort effect and period effect were assessed with relative risks”- Unless there is some detail missing in this sentence, there can only be one median date of birth among cases. The other part of the sentence is unclear.

The cohort and period effects were evaluated by relative risk, thus we selected different criteria as reference for cohort and period effects. And it's a typo for "median dates". We revised this statement as follows:

Line 139-143

"Birth cohort effect and period effect were assessed with relative risks. The median date of birth among cases were regarded as the reference cohort. Since cases aged at 35–85 years between 1998 and 2021 were selected, the range of birth cohort from 1913 to 1986 covered observations and further projections until 2030. The second and penultimate period effects were constrained to the reference for period."

Comment 4: For future projections, what assumptions were made for birth cohorts who were not present in the data but who would be in future populations?

Our prediction model was incorporated with age, period and cohort effects for different genders, age groups and immigration status, and we only consider deaths aged 35-85 as a few cases of two tails of the range, that is, the upper bound of birth cohort is 1986 (2021-35). We improve this point as follows:

Line 141-143

"Since cases aged at 35–85 years between 1998 and 2021 were selected, the range of birth cohort from 1913 to 1986 covered observations and projections until 2030."

Comment 5: Graphs are small and as such, are difficult to see; this is particularly so for the period graph where the x-axis should only extend to 2021 rather than 2040. Please check the titles are consistent with the graph headers (cohort and period effects are labelled differently in titles from graph headers). Most are labelled as "morality" instead of "mortality". In the projection graphs, the black line representing the predictive mean is difficult to see- the authors may want to consider a color of other than blue- or a different shade- for the 5-95% quantiles. Changing the aspect ratio of the projection graphs (i.e. make bit taller) may also help.

Appreciate the comments. All graphs have been revised i.e. illustration of APC estimates for each cancer have been merged as one figure; titles and x limits of them have been improved (Figure 1). Typos were revised, and heat color of quantiles for projections was used to make the lines of predictive means clear (such as Figure 2).

Comment 6: Lines 183-185 "For example, compared to no significant effects of immigration status for women on mortality rates of lung cancer by age (Figure 1b), the higher age effect for men who have stayed in Hong Kong for >10 years occurred after the age of 50 years and the lower age effect of men who had short stays occurred before the age of 62 years (Figure 1a)." Note that men and women are not being compared as such- it would be better to say that while there was no difference in lung cancer mortality rates by immigration status among women, male immigrants who remained in Hong Kong for >10 years had higher lung cancer mortality rates at ages above 50 years and those who arrived <=10 years had lower lung cancer mortality at ages below 62 years compared to local men. Other descriptions should follow this suggestion, making reference to the specific cancer mortality rates too.

Thank you very much for the comments! We revised these descriptions of results as follows:

Line 179-182

"...while relatively insignificant differences in lung cancer mortality rates by immigration status among females, male immigrants who remained in Hong Kong for >10 years had higher lung cancer mortality

rates at ages above 50 years and those who arrived ≤ 10 years had lower lung cancer mortality at ages below 62 years compared to local men.”

And line 183-190

“similar change of relative risks by birth cohort for locals and long-stay immigrants in lung, colon, liver and stomach cancers occurred before 1945, whereas significant differences of birth cohort effects between these two immigration groups occurred after 1960. Locals and long-stay immigrants in pancreatic and prostate cancer perform almost similar change of relative risks by birth cohort effects all the time. Short-stay immigrants who have stayed in Hong Kong for ≤ 10 years had more fluctuating relative risks affected by period effects before 2020 than those for locals and long-stay immigrants.”

Comment 7: Line 205- the text implies that prostate cancer could occur in either sex- please modify the text. Similarly in line 283.

The statements have been modified as follows:

Line 208-209

“Men will be at higher risk of mortality rates of cancers (excluding prostate cancer) than women in the future for all three age groups (all ages, young and older than 60 years).”

Line 292-293

“They are also compatible with the results in [4] that men suffer from higher risk of these types of cancer than women, excluding prostate cancer.”

Comment 8: As rates are presented by sex, why did the authors not consider reporting patterns for the most common cancers for each sex- so for women, breast, ovary, cervix?

Thank you very much. We did consider some other cancers and had to discard some of them since 1). Some of topic has been published, such as a paper which explore the relationship between the effect of immigration and breast cancer mortality

“Zhao S, Dong H, Qin J, Liu H, Li Y, Chen Y, Molassiotis A, He D, Lin G, Yang L. Breast cancer mortality in Chinese women: does migrant status play a role?. *Annals of Epidemiology*. 2019 Dec 1;40:28-34.”

2). The data was obtained from the Census and Statistics Department of Hong Kong, but a limited number of types of cancer was available for us right now. We have mentioned this point as a part of limitation and may consider it as an instructive potential of the research as follows:

Line 313-315:

“A few acceptable cases resulted in a limited type of cancer so that some common cancers such as ovary and cervix, were discarded.”

Comment 9: Lines 227-8 Cancer mortality rates increasing with age is known- what is relevant here is whether patterns are different among the immigrant and local populations. Consider removing this first sentence.

Thank you for your comments. We removed the first sentence and modified as follows:

Line 235-237

“While the changes in mortality rates by age for long-stay immigrants reached approximate harmony with those for locals, the changes in mortality rates by age for short-stay immigrants revealed clear differences with those for other two populations.”

Comment 10: Please check that the conclusions focus on the study objectives (differences in cancer mortality rates in different immigrant and local populations by age, cohort and period) and are consistent with the reported results.

Thank you very much for your comment. We revised our conclusion as:

Line 47-49:

“Findings underscore the effect of gender and immigration status in Hong Kong on mortality risks of cancers that immigrants for each type of cancer and gender will be at a higher mortality risk than locals.”

And Line 320-328:

“We conclude that immigrants, especially short-stay immigrants, had more pronounced fluctuations of mortality rates by age and of relative risks by cohort and period effects for six types of cancers than those of long-stay immigrants and locals. Men will be at a higher risk of mortality rates of six types of cancer than women in the future. Male immigrants who have stayed in Hong Kong for ≤ 10 years with colon cancer and male immigrants who have stayed in Hong Kong for > 10 years with pancreatic cancer would perform significant uptrend in the future, while other immigration groups for each type of cancer would continue to decline or remain relatively stable. Immigrants for each gender in Hong Kong would suffer from higher mortality risks of cancers than locals in the future.”

Comment 11: Please check the ethical statement- although the study uses routinely collected data, the study does involve human participants so the ethical statement may warrant revision.

It's truly a key issue. Thank you very much. The data was obtained from the Census and Statistics Department of Hong Kong, and we also review the point in some other literatures with similar issues on BMJ and BMJ Open for reference. Also, based on comments of editor, 'Patient and public involvement' statement has been removed to the end of the main text Methods section as follows:

“ Patient and Public Involvement
None. ”

Comments from the Reviewer #3, and author's response

Comment 1. It is not clear that the data are full-size data, I mean the number of deaths is a complete survey in the country. If the data were based on spontaneous report, it may make some biases in the analysis.

Some definitions of samples in details have been improved. The data is truly complete survey by Hong Kong government. We correct them as follows:

Line 110-111

“The data was extracted a routine census held by Hong Kong government as subjective errors caused by resampling can be neglected.”

Comment 2. Is not it necessary to describe the statistical significance of the model parameters. As the descriptions of the model development process are limited in this manuscript - mostly the manuscript is just quoting other references, I could not completely understand the parameter handling.

The primary objective of the descriptions of the models, especially APC model, was to depict a well-known non-identifiability problem thus there is only simple introduction of parameters (to describe the non-identifiability problem) and we mainly focused on the contributions of sex and immigration status, that is, observing the differences of age, period and cohort effect between genders and immigration groups. The motivation of the model was explained as:

Line 137-139

“We mainly focused on the contributions of sex and immigration status due to the non-identifiability problem that the effects of these three components are collinear with each other (denoted as period – age = cohort)”

Comment 3. For example, a sentence “... perform the most significant decline (Page 2, Line43), ... most significant uptrend (Page 2, Line 45) ...”, seems subjective and not based on strict mathematical basis.

We applied Mann-Kendall test to support our conclusions as follows:

Line 159:

“Mann-Kendall trend test was applied to verify the projection trend.”

Line 39-42:

“Decreasing trends ($p < 0.05$) or plateau ($p > 0.05$) of forecasting mortality rates of cancers occur for all immigration groups, except for increasing trends for short-stay male immigrants with colon cancer (16.77 deaths to 18.50 deaths/100,000 $p < 0.05$) and long-stay male immigrants with pancreatic cancer (17.87 deaths to 23.49 deaths/100,000 $p < 0.05$).”

Line 203-208

“Most of predictive trends for younger cases (< 60 years) and older cases (≥ 60 years) reach a consensus with those for all ages population, except for two phenomenon: 1.) mortality rates of lung cancer for men immigrants ≤ 10 that insignificant trend for all ages ($p\text{-value} > 0.05$) vs. decline for younger cases ($p\text{-value} < 0.05$) vs. increase for older cases ($p\text{-value} < 0.05$); 2.) mortality rates of liver cancer for men immigrants > 10 that decline for all ages ($p\text{-value} < 0.05$) vs. decline for younger cases ($p\text{-value} < 0.05$) vs. insignificant trend for older cases ($p\text{-value} > 0.05$).”

Comment 4. The future profile prediction is only based on the available data until 2022 with the covariate of age, cohort only. In fact, recent cancer therapy is progressing remarkably, and the future perspective of such progresses should be considered for the prediction. I think this is one limitation of the applied model here.

Thank you very much for your comment! We did consider such issue but realize the difficulty to quantify the progresses of therapies and techniques in the future. We revised the statements as follows:

Line 60:

“The future perspective of cancer therapies and techniques have not been considered.”

And line 315-316:

“Since the issue of quantification, the future perspective of cancer therapies and techniques have not been considered in the model of projection.”

VERSION 2 – REVIEW

REVIEWER	Rosenberg, Philip NIH, Bethesda, Biostatistics Branch
REVIEW RETURNED	26-Jun-2023

GENERAL COMMENTS	The breakdown by ages <60 versus >60 was nicely done and the details in the Supplement are outstanding. All my other comments were thoroughly addressed.
--

REVIEWER	Kane, Eleanor University of York, Health Sciences
REVIEW RETURNED	06-Jul-2023

GENERAL COMMENTS	Objectives do not specify why examining- and predicting- trends in mortality from these six cancers among immigrants in Hong Kong is of interest. While the authors have modified the abstract, the research is to examine whether mortality from lung, stomach, liver, colon, prostate and pancreatic cancer is different among migrants compared to the local population. Please state the objective more clearly. The authors should revise the paper to describe mortality rates among migrants relative to the local population. There is no need to describe that cancer mortality rises with age, that men have higher risk of mortality from these cancers than women or that there are pronounced fluctuations in rates for the short-term migrants (likely due to small numbers); please remove such text. It is also difficult to follow the description of future projections. It would be easier to follow if the focus was primarily on populations where the specific cancer mortality rate is likely to increase in the future; some secondary description of rates that remain stable is probably also of interest. Again, these descriptions should consider differences between migrants and local populations- and focus on those where migrants are at higher risk than locals (lower risk may be related to bias in the possible better health of short-term migrants). Rather than giving p-values for predicted trends in mortality rates over time, it may be clearer to give the change per year in the mortality rate with the 95% confidence interval. Also, reporting the expected mortality rates (and 95%CI) in the year 2030 compared to the last year of recorded data (2021) in the text for subpopulations with increasing rates and differences between migrants and locals will help the reader understand the potential changes. Please revise the abstract- as well as the results and discussion- in accordance with the above points. Other points to consider: It is unclear why period and cohort effects are presented as risks relative to a specific year (or years for period?). Why are mortality rates not used throughout? If there is a reason to use relative risks, please justify why relative risks are used and why risks are calculated relative to a specific year (rather than to locals in each cohort/period)?
---

	Graphs remain small and are still difficult to see; this is particularly so for the period graph- please use a different scale to the cohort section. Please use consistent y-scales across panels of graphs that are stratified by migrant status.
REVIEWER	Yano, Yoshitaka Kyoto Pharmaceutical University, Education and Research
REVIEW RETURNED	27-Jun-2023
GENERAL COMMENTS	No more comments.

VERSION 2 – AUTHOR RESPONSE

Comments from the Reviewer #1, and author's response

Comment1: The breakdown by ages <60 versus >60 was nicely done and the details in the Supplement are outstanding. All my other comments were thoroughly addressed.

We appreciate your constructive comments on different age strata. They make our paper more comprehensive and logical.

Comments from the Reviewer #2, and author's response

Comment 1: Objectives do not specify why examining- and predicting- trends in mortality from these six cancers among immigrants in Hong Kong is of interest. While the authors have modified the abstract, the research is to examine whether mortality from lung, stomach, liver, colon, prostate and pancreatic cancer is different among migrants compared to the local population. Please state the objective more clearly.

Thank you very much for your comment! As your comments previously, it's exactly what we are concentrating on how to balance word limit of abstract and concise explanation. We stated that we focused more on the effect of immigration on cancer mortality in the past and in the future. Thus, we revised our "objective" as follows:

Line 25-28

"To explore the relationship between immigration groups and cancer mortality, this study aimed to explore age, period, birth cohort effects and effects across genders and immigration groups on mortality rates of lung, pancreatic, colon, liver, prostate and stomach cancers and their projections."

And line 103-108

"...to evaluate the effect of immigration on cancer mortality in the past and future, we developed APC models specified by sex and migrant status to assess the effects of age, period, birth cohort, and of the length of stay in Hong Kong on the mortality risks of cancers. Additionally, we explore the projection of mortality rates for the locally born population and immigrants in Hong Kong who were

younger or older than 60 using a predictive model, taking into account age, period, and birth cohort effects as well.”

Comment 2: The authors should revise the paper to describe mortality rates among migrants relative to the local population. There is no need to describe that cancer mortality rises with age, that men have higher risk of mortality from these cancers than women or that there are pronounced fluctuations in rates for the short-term migrants (likely due to small numbers); please remove such text.

Thank you very much for your comment! We also revised and removed some descriptions like them. And revised in results as follows:

Line 202-206

“For all ages projection (Figure 2 & eFigure 2-6), as men will be at higher risk of mortality rates of cancers (excluding prostate cancer) than women in the future for all three age groups (all ages, young and older than 60 years, given the projected trends, immigrants for each gender, especially who have stayed in Hong Kong for > 10 years will suffer from higher mortality rates of cancer in each year than locals.”

Our primary purpose to these descriptions was also to state the effect of immigration on cancer mortality such that consistence seems also like an appropriate conclusion for the contrast. We compared the performance of age and gender effects among migrant status and highlighted approximately consistent conclusions in different migrant status that “cancer mortality rises with age” and “men have higher risk of mortality from these cancers than women”. Therefore, we stated that

Line 40-42

“Immigrants for each type of cancer and gender will be at a higher mortality risk than locals, as men will be at a higher risk of mortality from cancers than women in the future (excluding prostate cancer).”

For example, for each type of cancer, male immigrants will suffer from higher mortality than male locals, and male short-term (or long-term) immigrants will suffer from higher mortality than female short-term (or long-term) immigrants. Short-term migrants with pronounced fluctuations in rates partially resulted from different lifestyle, which is similar with what you mentioned in comment 3 and has been explained in the part of discussion.

Comment 3: It is also difficult to follow the description of future projections. It would be easier to follow if the focus was primarily on populations where the specific cancer mortality rate is likely to increase in the future; some secondary description of rates that remain stable is probably also of interest. Again,

these descriptions should consider differences between migrants and local populations- and focus on those where migrants are at higher risk than locals (lower risk may be related to bias in the possible better health of short-term migrants).

Thank you very much! Yes. The effect of immigration on cancer mortality is the key issue of the research. We modified some descriptions in abstract as well as in the main text as follows:

Line 40-47

“Immigrants for each type of cancer and gender will be at a higher mortality risk than locals, as men will be at a higher risk of mortality from cancers than women in the future (excluding prostate cancer). After 2021, decreasing trends ($p < 0.05$) or plateau ($p > 0.05$) of forecasting mortality rates of cancers occur for all immigration groups, except for increasing trends for short-stay male immigrants with colon cancer ($p < 0.05$, Avg +0.30 deaths/100,000 per annum from 15.47 to 18.50 deaths/100,000) and long-stay male immigrants with pancreatic cancer ($p < 0.05$, Avg +0.72 deaths/100,000 per annum from 16.30 to 23.49 deaths/100,000).”

And line 202-213

“For all ages projection (Figure 2 & eFigure 2-6), as men will be at higher risk of mortality rates of cancers (excluding prostate cancer) than women in the future for all three age groups (all ages, young and older than 60 years), given the projected trends, immigrants for each gender, especially who have stayed in Hong Kong for > 10 years will suffer from higher mortality rates of cancer in each year than locals. Monotone decreasing trends or plateau of forecasting occur for both genders and all immigration groups in cancers, except for increasing trends for male immigrants who have stayed in Hong Kong for ≤ 10 years with colon cancer ($p < 0.05$, Avg +0.30 deaths/100,000 per annum) from 15.47 deaths/100,000 (95% CI: 11.28, 19.66) in 2021 to 18.50 deaths/100,000 (95% CI: 2.31, 34.69) in 2030, and male immigrants who have stayed in Hong Kong for > 10 years with pancreatic cancer ($p < 0.05$, Avg +0.72 deaths/100,000 per annum) from 16.30 deaths/100,000 (95% CI: 14.38, 17.26) in 2021 to 23.49 deaths/100,000 (95% CI: 12.49, 34.49) in 2030.”

Comment 4: Rather than giving p-values for predicted trends in mortality rates over time, it may be clearer to give the change per year in the mortality rate with the 95% confidence interval. Also, reporting the expected mortality rates (and 95%CI) in the year 2030 compared to the last year of recorded data (2021) in the text for subpopulations with increasing rates and differences between migrants and locals will help the reader understand the potential changes.

Please revise the abstract- as well as the results and discussion- in accordance with the above points.

We appreciate your instructive comments! We revised all eTable 1-6 in Supplementary Materials and added the expected change per year and expected mortality rates in 2030 and the records of them in 2021 as follows:

Line 206-213

“Monotone decreasing trends or plateau of forecasting occur for both genders and all immigration groups in cancers, except for increasing trends for male immigrants who have stayed in Hong Kong for ≤ 10 years with colon cancer ($p < 0.05$, Avg +0.30 deaths/100,000 per annum) from 15.47 deaths/100,000 (95% CI: 11.28, 19.66) in 2021 to 18.50 deaths/100,000 (95% CI: 2.31, 34.69) in 2030, and male immigrants who have stayed in Hong Kong for > 10 years with pancreatic cancer ($p < 0.05$, Avg +0.72 deaths/100,000 per annum) from 16.30 deaths/100,000 (95% CI: 14.38, 17.26) in 2021 to 23.49 deaths/100,000 (95% CI: 12.49, 34.49) in 2030.”

Line 219-229

“Some particular cases occur in the projection of prostate cancer that young long-stay male immigrants (0.44 deaths/100,000, 95% CI: 0, 1.05) aged less than 60 will be at lower mortality rate than locals (0.69 deaths/100,000, 95% CI: 0, 1.42) in 2030 (eTable 6). Compared with other cancers and immigration groups, male immigrants who have stayed in Hong Kong for > 10 years with lung cancer would perform the most significant decline in predictive mean from 102.90 (95% CI: 98.14, 107.66) to 79.55 (95% CI: 47.46, 111.64) deaths per 100,000 population (Avg -2.34 deaths/100,000 per annum) (eTable 1), while the same immigration group with pancreatic cancer would indicate the most significant uptrend in each year of 16.30 (95% CI: 14.38, 17.26) and 23.49 (95% CI: 12.49, 34.49) deaths per 100,000 population in 2021 and 2030, respectively (Avg +0.72 deaths/100,000 per annum) (eTable 4).”

P-values are still considered in the text due to other comments, and it can be a support for results. For example, the trend of mortality for male locals who would suffer from pancreatic cancer would be insignificant ($p\text{-value} > 0.05$), even though a slightly increasing trend performed from 2021 (11.97) to 2030 (14.02). So we used p-values as the one to show significant and insignificant trends.

Other points to consider:

Comment 5: It is unclear why period and cohort effects are presented as risks relative to a specific year (or years for period?). Why are mortality rates not used throughout? If there is a reason to use relative risks, please justify why relative risks are used and why risks are calculated relative to a specific year (rather than to locals in each cohort/period)?

Thank you very much! This is a really important comment. We didn't explain some details of the model (APC) since we hoped to refine our introduction and focused more on results, but it truly seems to result in some misunderstandings. Due to the “non-identifiability problem”, a consensus of APC in most of application is to select reference points for the period and cohort effects to evaluate the effect of three components independently. There is always one term with the rate dimension (usually age), but it must refer to specific reference values for other variables (period and cohort in the model). For the other variables, report the relative risk to the reference point

odds-ratios, e.g. Age-effects as rates for the reference cohort, cohort and period effects as relative risks to the reference cohort.

The estimated age-specific mortality rates in the 1965-cohort.

The cohort rate-ratio relative to the 1965-cohort.

The period rate-ratio taken as a residual relative risk.

To support our model default, we added two citations which are the theorem and its application:

“Yang Y, Land KC. Age-period-cohort analysis: New models, methods, and empirical applications. Taylor & Francis; 2013.”

“Robertson C, Gandini S, Boyle P. Age-period-cohort models: a comparative study of available methodologies. Journal of clinical epidemiology. 1999 Jun 1;52(6):569-83.”

And revised some introduction and as follows:

Line 143-145

“Birth cohort effect and period effect were assessed with relative risks to evaluate the effect of three components. The median year of birth among cases were regarded as the reference cohort [35,36].”

Comment 6: Graphs remain small and are still difficult to see; this is particularly so for the period graph- please use a different scale to the cohort section. Please use consistent y-scales across panels of graphs that are stratified by migrant status.

Thank you very much. We should have considered some issues of graphic layout. We revised our figures in main text such as Figure 1 & 2 to make them easy to recognize. Cohort and period sections are in the continuous scale since death cases aged at 35–85 years between 1998 and 2021 were selected and thus the range of birth cohort was from 1913 to 1986. If we use consistent y-scales, such as the same range of mortality from 1 to 600 (as well as range of relative risk from 0.1 to 60) for “male lung” and “female lung”, graphs for “female lung” would be more difficult to see the differences between migrant status. Due to the “non-identifiability problem”, what we focused more on the APC figures were the trends and differences between migrant status instead of quantities, which have also reflected the descriptions in part of results that there are only descriptions of trends, and we pay more attention on the quantities in projections (that’s also another reason for the description of future projection in comment 3 above).

Comments from the Reviewer #3, and author’s response

Comment 1. No more comments.

It’s indeed necessary for more strict statistical tests instead of description. Appreciate your comments which make our paper more relevant.

VERSION 3 – REVIEW

REVIEWER	Kane, Eleanor University of York, Health Sciences
REVIEW RETURNED	19-Aug-2023

GENERAL COMMENTS	Thank you for addressing my comments. There are a few comments remaining that require attention. One comment is with respect to the scales of the graphs- to clarify there are two issues to be resolved here. 1) The first is the use of the same scaling on the x-axis for the period as for the cohort effects. While both are of course in years, the range of values are different, making the cohort with its wider range much easier to see than the period graphs with its smaller one. Different scales can be used for the cohort and period graphs- please increase the scaling for the period graphs. 2) The other is consistency of the y-axis ranges within each of Figures 3-5. The y-axis range should be the same across all three graphs for males in each figure; i.e in Figure 3, y-values should range from 0 to 150; in Figure 4, from 0 to 60, etc. The same should be applied to the graphs for the females. This will make the visual appearance of the gradients and confidence interval ranges consistent across the graphs that are being compared. The other comment is with respect to the phrasing that males have higher cancer mortality than women. The primary question of whether immigrants have higher cancer mortality than locals has been addressed. What remains unclear is whether the differences in cancer mortality between immigrants and locals is different for men than it is for women (comparisons of male to female immigrants (or locals) may be what is expected given known sex differences). Did the authors test for interactions between sex and immigration status to aid the interpretation? The text as it stands remains unclear and the authors should rephrase to clarify whether the data shows that differences in cancer mortality between male immigrants and male locals is a different pattern to that between female immigrants and female locals.
--

VERSION 3 – AUTHOR RESPONSE

Comments from the Reviewer #2, and author's response

Comment 1: One comment is with respect to the scales of the graphs- to clarify there are two issues to be resolved here.

1) The first is the use of the same scaling on the x-axis for the period as for the cohort effects. While both are of course in years, the range of values are different, making the cohort with its wider range much easier to see than the period graphs with its smaller one. Different scales can be used for the cohort and period graphs- please increase the scaling for the period graphs.

2) The other is consistency of the y-axis ranges within each of Figures 3-5. The y-axis range should be the same across all three graphs for males in each figure; i.e in Figure 3, y-values should range from 0 to 150; in Figure 4, from 0 to 60, etc. The same should be applied to the graphs for the

females. This will make the visual appearance of the gradients and confidence interval ranges consistent across the graphs that are being compared.

Thank you very much for your instructive comments all the time! We reviewed all figures and revised them as 1) wider range for period effect in Figure 1&2 and eFigure 1(a)-(e) Supplementary Materials; 2) consistent range of y-axis for each gender in Figure 3-5 and eFigure 2-6 Supplementary Materials.

Comment 2: The other comment is with respect to the phrasing that males have higher cancer mortality than women. The primary question of whether immigrants have higher cancer mortality than locals has been addressed. What remains unclear is whether the differences in cancer mortality between immigrants and locals is different for men than it is for women (comparisons of male to female immigrants (or locals) may be what is expected given known sex differences). Did the authors test for interactions between sex and immigration status to aid the interpretation? The text as it stands remains unclear and the authors should rephrase to clarify whether the data shows that differences in cancer mortality between male immigrants and male locals is a different pattern to that between female immigrants and female locals.

Thank you very much for your comment! Issues related to interactions are also considered. Basically, with some statistical tests such as ANOVA and observed results from tables in Supplementary Materials, the interaction between gender and immigration status is seems approximately significant for each type of cancer in each year (excluding prostate cancer), that is, the differences in cancer mortality between immigrants and locals is different for men than it is for women. We applied Friedman test for each to support our findings as follows:

Line 164-165

“Friedman’s Two-Way Analysis of Variance was applied to test interactions between gender and immigration groups for each year.”

And line 204-210

“For all ages projection (Figure 2 & eFigure 2-6), as men will be at higher risk of mortality rates of cancers (excluding prostate cancer) than women in the future for all three age groups (all ages, young and older than 60 years) and approximately significant interactions between gender and immigration groups emerge for each type of cancer in each year ($p < 0.05$), given the projected trends, immigrants for each gender, especially who have stayed in Hong Kong for > 10 years will suffer from higher mortality rates of cancer in each year than locals.”

VERSION 4 – REVIEW

REVIEWER	Kane, Eleanor University of York, Health Sciences
REVIEW RETURNED	04-Sep-2023

GENERAL COMMENTS	Thank you for amending the graphs- these all now look much better. Thank you also for including information on the tests for interaction by gender and immigration status within each period year. Sorry to reiterate this, but there has been no attempt to change statements regarding higher cancer mortality among men than women. As stated in previous reviews, this is known- the question
--

	here is whether there are sex-specific immigrant groups who are at a greater risk of cancer mortality than expected. The phrases are:  1) Abstract lines 41-42 "... as men will be at a higher risk of mortality from cancers than women in the future (excluding prostate cancer)." 2) Results lines 204-206 "For all ages projection (Figure 2 & eFigure 2-6), as men will be at higher risk of mortality rates of cancers (excluding prostate cancer) than women in the future for all three age groups (all ages, young and older than 60 years)..." 3) Conclusion lines 332-333 "Men will be at a higher risk of mortality rates of six types of cancer than women in the future." Following each of these statements, the authors do describe that mortality from colon cancer and pancreatic cancer are predicted to increase in male short- and long-stay immigrants respectively. As such, please remove the phrases listed above for better clarity of the study's findings.
--	--

VERSION 4 – AUTHOR RESPONSE

Comments from the Reviewer #2, and author's response

Comment 1: Thank you for amending the graphs- these all now look much better.

Thank you also for including information on the tests for interaction by gender and immigration status within each period year.

Sorry to reiterate this, but there has been no attempt to change statements regarding higher cancer mortality among men than women. As stated in previous reviews, this is known- the question here is whether there are sex-specific immigrant groups who are at a greater risk of cancer mortality than expected.

The phrases are:

- 1) Abstract lines 41-42 "... as men will be at a higher risk of mortality from cancers than women in the future (excluding prostate cancer)."
- 2) Results lines 204-206 "For all ages projection (Figure 2 & eFigure 2-6), as men will be at higher risk of mortality rates of cancers (excluding prostate cancer) than women in the future for all three age groups (all ages, young and older than 60 years)..."
- 3) Conclusion lines 332-333 "Men will be at a higher risk of mortality rates of six types of cancer than women in the future."

Following each of these statements, the authors do describe that mortality from colon cancer and pancreatic cancer are predicted to increase in male short- and long-stay immigrants respectively. As such, please remove the phrases listed above for better clarity of the study's findings.

Thank you very much for your comments! We ever considered that a consistent conclusion was also meaningful and may be the support for the credibility of our findings so that we stated the result, which was comparable with a well-known finding, that "Men will be at a higher risk of mortality rates of six types of cancer than women in the future". Therefore, inspired by your instructive comments, we

removed the phrases listed above, especially those in part of “Results”, and also expounded more in the part of “Discussion” as follows:

Line

“Men will be at higher risk of mortality rates of cancer than women, regardless of immigration status. They are also compatible with the results in [4] that men suffer from a higher risk of these types of cancer than women, excluding prostate cancer. Furthermore, new immigrant women will be at lower risk than local women, even though long-stay immigrants will suffer from higher mortality rates than locals in the future. Potential interpretations could be consistent with those for birth cohort effects, as age and period effects are considered as confounders of cohort effect.”

VERSION 5 – REVIEW

REVIEWER	Kane, Eleanor University of York, Health Sciences
REVIEW RETURNED	06-Sep-2023
GENERAL COMMENTS	Thank you for addressing my comments- I have no further suggestions to make.

VERSION 5 – AUTHOR RESPONSE